# A Cross-Sectional Analysis of the Stigma Surrounding Type 2 Diabetes in Colombia

**DOI:** 10.3390/ijerph182312657

**Published:** 2021-12-01

**Authors:** Victor Pedrero, Jorge Manzi, Luz Marina Alonso

**Affiliations:** 1Nursing Faculty, Universidad Andrés Bello, Santiago 8370146, Chile; 2School of Psychology, Pontificia Universidad Católica de Chile, Santiago 7820436, Chile; jmanzi@uc.cl; 3Department of Public Health, Universidad del Norte, Barranquilla 081007, Colombia; lmalonso@uninorte.edu.co

**Keywords:** social stigma, type 2 diabetes mellitus, self-management

## Abstract

Type 2 diabetes is a global epidemic, and many people feel stigmatized for having this disease. The stigma is a relevant barrier to diabetes management. However, evidence in this regard is scarce in Latin America. This study aimed to analyze the level of stigma surrounding type 2 diabetes in the Colombian population and its relationships with sociodemographic, clinical, psychosocial variables and behaviors related to management of the disease (self-management behaviors). This cross-sectional study included 501 Colombian adults with type 2 diabetes. We estimated the relation between stigma and selected variables through linear regression models. Additionally, we analyzed the mediator role of psychosocial variables in the relationship between stigma and self-management behaviors through structural equation models. A total of 16.4% of patients showed concerning levels of stigma. The time elapsed since diagnosis (β = −0.23) and socioeconomic status (β = −0.13) were significant predictors of the level of stigma. Stigma was negatively correlated with self-efficacy (β = −0.36), self-esteem (β = −0.37), and relationship with health care provider (β = −0.46), and positively correlated with stress (β = 0.23). Self-efficacy, self-esteem, and the relationships with health care providers had a mediation role in the relationship between stigma and self-management behaviors. These variables would be part of the mechanisms through which the perception of stigma harms self-management behaviors. The stigma of type 2 diabetes is frequent in the Colombian population and negatively associated with important aspects of disease management.

## 1. Introduction

According to the latest report from the International Diabetes Federation, 88.8 million people have type 2 diabetes in Latin America, and this number is expected to increase to 108 million people by 2045 [1]. In Colombia, the prevalence of type 2 diabetes is 8.5%, which is equivalent to more than 2 million people [2,3]. However, many people with diabetes do not know their diagnosis. Barengo et al. [4] estimate that the prevalence of undiagnosed diabetes in Barranquilla Colombia was 5.1% in a general population sample. Risk screening systems are crucial to address this situation [4]. Recent progress in this area has been made; an example is the use of artificial intelligence algorithms to develop locally adapted screening tools [5,6]. These advances make it possible to act promptly and reduce the complications of this disease.

Data on the degree of metabolic compensation presented by patients with type 2 diabetes in Latin America are limited [7]. According to the DISCOVER study, the mean glycated hemoglobin (HbA1c) level in the region is 8% [8]. Other sources suggest that depending on the country, between 3.5% and 54% of patients in the region have HbA1c levels below 7% [2,3]. Particularly in Colombia, only 13% of patients show adequate metabolic compensation [2,3].

Similar to other international organizations, the Latin American Diabetes Association recommends focusing on psychosocial aspects to achieve proper disease management and control [9,10,11]. The stigma surrounding diabetes has emerged as a new factor to consider in this area [12]. Studies conducted in Australia and the United States have shown that perceived stigma is frequent among patients with type 2 diabetes, and that between 17.1% and 52% of this population feels stigmatized for having the disease [13,14,15]. In Latin America, the study of this phenomenon is incipient. In 2013, an international study exploring the Mexican population with diabetes indicated that 13.9% of patients in this group experienced stigma associated with the disease [15].

Stigma has been traditionally defined in reference to a characteristic that discredits those with that characteristic considering its social connotation [16]. In this case, a type 2 diabetes diagnosis is the characteristic, and the social connotation is reflected by the stereotypes associated with the disease, including beliefs that those with diabetes are responsible for having the disease, are unable to carry out certain tasks of daily living, or have a high probability of dying [17]

The stigma surrounding diabetes has important negative effects on both the metabolic compensation and quality of life of people with the disease [18,19,20,21]. However, the mechanisms involved in this relationship still require investigation. Evidence in this area has shown that stigma can negatively affect psychosocial and behavioral aspects that are key to achieving good disease management [19,20,22,23,24].

From the psychosocial point of view, stigma has been negatively correlated with self-efficacy and self-esteem and positively correlated with stress among people who suffer from the disease [20,23,25,26]. Additionally, relationships with health care providers can be affected by perceived stigma, which may lead patients to avoid medical consultations [20,27]. Diabetes stigma has also been linked to less engagement in diabetes self-management behaviors (e.g., adherence to pharmacological treatments and lifestyle changes) [19,23,26,28].Some psychosocial variables, such as those mentioned above, may have a mediation role in the relationship between stigma and self-management behaviors [12,20,23,26,29].

Clinical and sociodemographic factors may influence the level of stigma experienced by patients with diabetes [13,20,30,31]. Patients with diabetes-related complications, who use insulin, and have been recently diagnosed may experience higher levels of stigma. Likewise, younger individuals and those who have a lower socioeconomic status are also more vulnerable to the diabetes stigma [13,20,30,31].

Research on the stigma surrounding diabetes is scarce in Latin America, even though the disease is expanding [2,7,20]. Elucidating the prevalence, mechanisms, and factors associated with the stigma will increase awareness of this phenomenon in the region, guide possible interventions in this area, and contribute to international evidence on the topic. Such advances may contribute to better and more focused health care provision to people with diabetes.

The present research aimed to analyze the stigma surrounding type 2 diabetes in the Colombian population suffering from the disease. Particularly, we explored the following aspects: (I) the prevalence of diabetes stigma in the sample; (II) the relationship between diabetes stigma and sociodemographic and clinical factors; (III) the connection between stigma and other psychosocial factors (self-efficacy, self-esteem, psychological stress, and relationships with health care providers) and behavioral aspects (self-management behaviors) relevant to diabetes management; and (IV) the mediating roles of psychosocial factors in the relationship between stigma and self-management behaviors.

According to the background, we have three main hypotheses. 

**Hypothesis** **1** **(H1).***Those who are younger, report a lower socioeconomic status, have been recently diagnosed, use of insulin, and have disease-related complications would be more susceptible to stigma*. 

**Hypothesis** **2** **(H2).***Regarding the psychosocial variables, a higher perception of stigma would be associated with lower self-efficacy and self-esteem, a worse perception of the patient–provider relationship, higher psychological stress, and a lower presence of self-management behaviors*. 

**Hypothesis** **3** **(H3).***Finally, we expect that self-esteem, self-efficacy, psychological stress, and relationship with the health provider would mediate the association between stigma and self-management behaviors*.

## 2. Materials and Methods

### 2.1. Design and Data Collection

This study had an observational cross-sectional design. The inclusion criteria for the sample were a medical diagnosis of type 2 diabetes and an age older than 18 years. Exclusion criteria were not applied. The final sample size of this study was 501 participants (Table 1), which was adequate to satisfy the requirements of the mediation analysis (see details in the Section 2.3) [32,33].

The participants were recruited from three zones of the city of Barranquilla (southwestern zone, southeast zone, and metropolitan zone), Colombia, during 2019. The local health authority provided the contact information of possible participants, who were contacted by telephone by the research team. If a patient expressed interest in participating, a research assistant visited the patient at home. During this visit, the informed consent form was explained to the participant. Once the participant signed the consent form, a survey was administered. Each enrolled patient was asked to contact new participants who met the study’s inclusion criteria, who were then invited to participate following the same protocol described.

This study was approved by the ethics committee of Universidad del Norte in Colombia (assessment opinion no. 197).

### 2.2. Measurements

#### 2.2.1. Sociodemographic and Clinical Variables

The participants answered questions about their sex (male/female), age (years), and subjective socioeconomic status, the latter of which was measured using the MacArthur ladder [34]. This indicator asks participants to place themselves on a drawn 10-rung ladder. The instructions state that the people with the least money, less education, and the worst jobs in society are on rung 1, and the people with the most money, most education, and best jobs are on rung 10. This measure has shown good correlations with objective indicators of social position and health indicators [35].

Regarding clinical indicators, the following self-report measures were included: time since diabetes diagnosis, insulin use, and presence of diabetes-related complications (ulcers, ophthalmological problems, and kidney problems). This last variable was dichotomized (no complications/some complications).

#### 2.2.2. Stigma Surrounding Type 2 Diabetes

We used a Spanish version of the Type 2 Diabetes Stigma Assessment Scale (DSAS-2) [14]. The DSAS 2 has 19 items scored on a 5-point Likert-type response scale (strongly agree–strongly disagree). The score of this scale ranges from 19 to 95 points, where a higher score indicates a higher level of stigma. The reliability of this instrument in our sample was α = 0.76.

#### 2.2.3. Self-Efficacy

Patients answered 7 items of the Spanish version of the self-efficacy scale for diabetes developed by Lorig, et al. [36] (i.e., How confident do you feel that you can follow your diet when you have to prepare or share food with other people who do not have diabetes?). This instrument measures patients’ perceptions of their ability to manage the disease. Each item has a 10-point response scale, and higher scores indicate a higher level of self-efficacy. The observed reliability in this study was α = 0.81.

#### 2.2.4. Self-Esteem

We used only one item to measure this variable: Indicate the extent to which you agree with the following statement: I have high self-esteem. This item has a 7-point response scale, and higher scores indicate greater perceived self-esteem. Robins, et al. [37] analyzed the validity and reliability of this question. Regarding reliability, they found a test–retest correlation of 0.76. In addition, this question showed a correlation of r = 0.93 with the Rosenberg Self-Esteem Scale [38], which is one of the most commonly used instruments to measure self-esteem [39].

#### 2.2.5. Psychological Stress

The participants completed the Spanish version of the 5-item Problem Areas in Diabetes (PAID-5) scale [40,41]. This instrument measures the degree of psychological stress that patients experience as a result of living with diabetes. The PAID-5 addresses 5 areas that may be problems for patients with diabetes (e.g., coping with complications of diabetes). Each item is answered on a 5-point Likert scale (not a problem–serious problem). Higher scores indicate greater psychological stress. The reliability of this instrument was α = 0.75.

#### 2.2.6. Self-Management Behaviors

Self-management behaviors were measured using 7 items of the Diabetes Self-Management Questionnaire (DSMQ) [42]. Each item is answered on a 4-point Likert scale (does not apply to me–applies to me very much). The selected items were chosen based on the magnitude of their factor loading in the original study and their correlations with HbA1c levels. Thus, we utilized 1 item for dietary control (i.e., Occasionally, I eat lots of sweets or foods rich in carbohydrates); 2 items for physical activity (e.g., I avoid physical activity, although it would improve my diabetes); 2 items for glucose management (e.g., I tend to forget to take or skip my diabetes medication); and 2 items for health care use (e.g., I tend to avoid diabetes-related doctor appointments). Overall scale showed a reliability of α = 0.65 in the sample.

#### 2.2.7. Relationships with Health Care Providers

The variable relationships with health care providers was measured with the Patient Satisfaction Questionnaire (PSQ) [43]. This instrument has 5 items measuring (1) patient’s satisfaction with how his or her needs were addressed, (2) his or her active involvement in the interaction, (3) information received, (4) emotional support received, and (5) the interaction in general. Each of the questionnaire items is scored on a scale of 1 to 10, and higher scores indicate greater satisfaction. The reliability of this instrument was α = 0.99.

### 2.3. Analysis

We descriptively analyzed the presence of stigma among the participants based on the DSAS-2 scores, including both the total score and the scores on each of the items composing the scale.

To examine the relationships between diabetes stigma and sociodemographic (i.e., sex, age, and socioeconomic status) and clinical factors (i.e., the time since diabetes diagnosis, insulin use, and the presence of diabetes-related complications), we calculated mean differences and correlations, and performed a multiple regression model. In the regression model, the dependent variable was stigma, and the independent variables were sociodemographic and clinical variables. First, only the sociodemographic variables were introduced in the regression analysis, and then the clinical variables were added.

To analyze the connection between the level of stigma and other psychosocial factors (self-efficacy, self-esteem, psychological stress, and relationships with health care providers) and behavioral (self-management behaviors) aspects relevant to diabetes management, we used correlations and linear regression models. We ran five regression analysis, one model for each psychosocial and behavioral factor (dependent variable). In all cases stigma was an independent variable. All these models were controlled for the sociodemographic and clinical variables, and standardized coefficients were estimated. All analyses considered a bootstrap estimation of standard errors, and a significance level of *p* < 0.05 was adopted. These estimations were performed in IBM SPSS Statistics for Macintosh Version 25.0 (Armonk, NY, USA).

To evaluate if the psychosocial variables included in this study were part of the mechanism involved in the relationship between stigma and self-management behaviors, we carried out a mediation analysis using structural equation models (SEM). Mediation analysis and SEM are highly popular in Social Sciences and have become increasingly relevant in epidemiology and public health [44,45].

In theory, the mediation specifies a chain of connections in which an exposure variable (i.e., stigma) affects a mediating variable (i.e., psychosocial variables), which in turn affects an outcome (i.e., self-management behaviors) (Figure 1) [44,45]. Technically, the mediation analysis corresponds to a series of regression models, which decomposes the exposure–outcome relationship into a direct and indirect effect through the mediator variable [44,45]. In our case, the direct effect represents the relationship between stigma and self-management behaviors when adjusted for the mediator (c’). In contrast, the indirect effect represents the effect of stigma on self-management through the mediator. The indirect effect can be estimated as a product of the regression coefficients representing the relationship between (i) stigma and psychosocial variables (a); (ii) psychosocial variables and self-management (b). The relative size of the mediated effect can be determined as a proportion of the total effect (the sum direct and indirect effect) [44,45].

The SEM approach allows us to perform a mediation analysis using latent or non-observable variables [46]. This strategy has some advantages as an explicit assessment of a measurement error, which is essential when using measurement instruments [46]. Each variable in the mediation analysis was modeled as a latent factor (circles in Figure 1) using the questionnaire’s items as indicators, except for self-esteem, which was measured with only one item. We conducted four mediation analyses using SEM to evaluate the relationship between stigma and self-management behaviors, one for each mediator variable of interest.

SEM usually demand big samples sizes, especially when each latent variable is modeled from many indicators, as in our case [47]. To address this problem we used parcels [47]. A parcel corresponds to the sum or average of 2 or more items representing a certain latent variable (squares in Figure 1) [47]. To construct the parcels, we carried out a 3-stage process. First a 1-factor model was fitted for each psychosocial or behavioral variable of interest. Second, in each case the items were grouped according to the magnitude of their factor loadings. Thus, the parcels were composed of items with factor loadings of different magnitudes. Third, the items constituting each of the parcels, usually between 3 and 5, were averaged. These parcels were introduced in the model as indicators of their respective latent variables. For example, in the case of stigma surrounding diabetes, a latent variable was modeled from 4 parcels (3 parcels with five items and 1 with four). The relationship between a parcel and a latent variable was represented as a factor loading (λ).

To assess the fit of each SEM, we used 3 commonly recommended goodness-of-fit indicators [46]: the root mean square error of approximation (RMSEA), comparative fit index (CFI), and standardized root mean square residual (SRMR). The RMSEA is expected to be ≤0.08, the CFI is expected to be ≥0.95, and the SRMR is expected to be ≤0.08. In each case, direct and indirect effects were estimated in a standardized manner, and a level of significance of *p* < 0.05 was considered for each effect. These estimations were computed using Mplus 8 [48].

Overall, 12 cases of missing data for different variables were detected. In the descriptive analyses, we considered the total data available for each variable. In the regression analysis, only complete data were used, whereas in the mediation analysis, all the available data were used, employing a full information maximum likelihood algorithm. This algorithm allows the use of all the data available for each variable present in the analysis [49].

## 3. Results

### 3.1. Level of Stigma

The mean score obtained on the DSAS-2 was 49.79 and standard deviation (SD) 7.11, with scores ranging from 25 to 74 points (Table 2). The original authors of the DSAS-2 proposed that scores greater than 1 SD above the mean total score indicate problematic levels of stigma [14]. Our results show that 16.4% of patients are in this situation.

Although the mean score on the DSAS-2 for the Colombian population is similar to that previously reported in Australia [14], four items showed markedly higher levels of agreement in the Colombian study than in the Australia study. In our sample, 83.3% of patients agreed or strongly agreed that a negative stigma applies to type 2 diabetes being a lifestyle disease (Table 3). A total of 60.9% think that health professionals have negative judgments about them for having the disease, 88.5% believed that health professionals thought that people with diabetes do not know how to take care of themselves, and 40.6% of the respondents experienced feelings of guilt for having the disease.

### 3.2. Factors Associated with Stigma

To test H1, different analyses were carried out. The descriptive analysis with sociodemographic variables shows that women (M = 50.37, SD = 7.06) have a significantly higher perceived stigma than men (M = 48.81, SD = 7.12)), t(378.74) = −2.36, *p* = 0.025). A negative correlation was observed between the presence of stigma and perceived socioeconomic status (r = −0.15, *p* = 0.001). Age did not show a significant association with the level of stigma (r = −0.09, *p* = 0.06).

Regarding the clinical variables, a negative correlation was observed between stigma and the years since diagnosis (r = −0.27, *p* < 0.001). Contrary to expectations, our data show that patients who used insulin had significantly lower levels of stigma (M = 48.64, SD = 7.08) than those who did not use this therapy ((M = 50.43, SD = 7.06), t(360.07) = 2.69, *p* = 0.009). This finding may be partly explained by the positive correlation between insulin use and the time elapsed since diagnosis (r = 0.31, *p* < 0.001). No significant differences were observed between those who reported having diabetes-related complications (M = 49.73, SD = 7.19) and those who reported having no complications ((M = 49.85, SD = 7.19), t(219.26) = 0.166, *p* = 0.861).

Table 4 shows the results of the multiple regression models. Model 1 shows that sociodemographic variables explain 4.2% of the variance in stigma (R2 = 0.042, F(3,487) = 7.11, *p* < 0.01). In this model, age and socioeconomic status were significant variables. The second model, which incorporates the three explored clinical variables, explains 10.1% of the variance (R2 = 0.101, F(6,484) = 9.04, *p* < 0.01), suggesting that the clinical variables are responsible for 5.9% of the variance in stigma. In model 2, socioeconomic status remained significant (β = −0.13, *p* = 0.002), in addition to the time elapsed since diagnosis (β = −0.23, *p* = 0.001), whereas age became nonsignificant (β = −0.08, *p* = 0.138), which may be due to the correlation between age and the years since diagnosis (r = 0.18, *p* < 0.001). These results partially confirmed H1.

### 3.3. Relationship between Stigma and Psychosocial and Behavioral Aspects

The results of the analysis related to H2 are presented in Table 2 and Table 5. The correlation analysis showed that stigma is significantly related to all psychosocial and behavioral variables (Table 2). Specifically, moderate and negative correlations were identified between the level of stigma and self-efficacy (r = −0.37, *p* < 0.001), self-esteem (r = −0.36, *p* < 0.001), and satisfaction with the care received (r = −0.45, *p* < 0.001). The correlation with self-management was also negative, but weaker (r = −0.24, *p* < 0.001). For stress, the correlation was positive and weak (r = 0.20, *p* < 0.001), indicating that higher levels of stigma are associated with higher levels of stress.

The regression models that consider sociodemographic and clinical factors and the presence of stigma as explanatory variables account for 17% of the variability in self-efficacy (R2 = 0.17, F(7,483) = 14.279, *p* < 0.001); 18% of the variability in self-esteem (R2 = 0.18, F(7,483) = 15.145, *p* < 0.001); 10% of the variability in diabetes-related stress (R2 = 0.1, F(7,482) = 7.293, *p* < 0.001); 10% of the variability in self-management (R2 = 0.1, F(7,482) = 7.55, *p* < 0.001); and 22% of the variability in the relationships with health care providers (R2 = 0.22, F(7,483) = 19.39, *p* < 0.001). In all cases, consistent with H2, stigma showed the highest standardized regression coefficient, suggesting that this is the most relevant explanatory variable in each case (Table 5).

The variable relationships with health care providers was the most affected by stigma. A change of 1 standard deviation in the stigma score (approximately 7 points) is associated with a change in almost half of 1 standard deviation (approximately 4 points) in the level of satisfaction with the relationship with health care providers (β = −0.46, *p* < 0.001). Likewise, an increase of 1 standard deviation in perceived stigma led to a decrease of approximately 40% of 1 standard deviation in perceived self-efficacy (β = −0.36, *p* < 0.001) and self-esteem (β = −0.37, *p* < 0.001). The variables least affected by stigma were perceived stress and adherence to self-management behaviors.

### 3.4. Mediation Analysis

The results of the SEM models used to test H3 are shown in Figure 2. In each diagram, it is possible to observe the number of parcels (P) built for each latent variable, goodness of fit indices, and size of the mediation effects. All SEMs showed a good fit to the data. The mediation analysis shows that, except for stress (β = −0.017, *p* = 0.184), all the psychosocial variables studied significantly mediate the relationship between stigma and self-management (Figure 1). For self-efficacy, we observed a mediating effect of greater magnitude (β = −0.281, *p* < 0.001), which accounted for 67.47% of the total effect of stigma on self-management behaviors, implying that a greater experience of stigma precipitates a decrease in self-efficacy levels, potentially impairing the ability of patients to engage in self-management behaviors. Self-esteem (β = −0.121, *p* < 0.001) and relationships with health care providers (β = −0.181, *p* < 0.001) also showed a significant mediating effect but with a lower magnitude. Moreover, 29.09% of the total effect of stigma on self-management behavior was found to depend on self-esteem, whereas this percentage was 43.5% for relationships with health care providers. These results partially supported H3.

## 4. Discussion

Our results indicate that 16.4% of patients experience concerning levels of stigma. Regarding the hypotheses, the H1 was partially supported. When analyzing the factors that may affect perceived stigma, a more recent diagnosis and a lower socioeconomic level were found to be associated with higher levels of stigma. However, we did not find a relationship between stigma and disease-related complications or insulin use. As predicted in H2, stigma was linked to lower perceived self-efficacy and self-esteem, greater psychological stress, worse perceived relationships with health care providers, and a lower tendency to engage in self-management behaviors. The mediation hypothesis (H3) was partially fulfilled. The analysis showed that part of the effect of stigma on self-management behaviors depends on self-efficacy, self-esteem, and relationships with health care providers. However, psychological stress was not a significant mediator.

The prevalence of stigma surrounding diabetes in this study is close to that found in other parts of the world. Browne, Ventura, Mosely and Speight [14] reported that 19.3% of a sample of more than 1000 patients in Australia felt stigmatized for having type 2 diabetes. In turn, the results of the DAWN2 study, which analyzed a population from 17 countries, showed that the overall prevalence of stigma surrounding diabetes was 17.1% [15]. However, the results of both studies should be interpreted with caution, as while the study by Browne, Ventura, Mosely and Speight [14] used the DSAS-2 scale, the DAWN2 study only used 1 item to measure stigma (i.e., I have been discriminated against because I have diabetes).

One aspect that contributes to stigma surrounding this disease is the stereotypes associated with it. These include stereotypes that emphasize individual responsibility, given that diabetes is considered a lifestyle disease [50]. Although lifestyle is a factor contributing to the onset of type 2 diabetes, the role played by the social determinants of health (e.g., socioeconomic status and access to health services, among others) cannot be overlooked [51]. In our study, 83.6% of patients indicated that they agree that diabetes has a negative stigma as it is associated with lifestyle, which is markedly higher than the previously reported rate of 44.6% [14]. The stereotypes emphasizing individual responsibility can be internalized by those who suffer from the disease and expressed in the form of certain emotions (e.g., guilt), which has been associated with worse health indicators [52]. In our study, 45% of the participants stated that they felt guilty for having the disease, which is substantially higher than the previously reported rate of 30.5% [14]. This finding may imply cultural differences around stereotypes and how the stigma surrounding diabetes is conceptualized in different regions.

Our results show that patients experience higher levels of stigma in the context of health care than in other environments, such as work or family, which may be attributed to the low visibility of the disease in these contexts. A total of 60.9% of patients reported feeling judged by health care providers, which is far higher than the rate reported in previous reports indicating that between 18% and 23.8% of patients were in this situation [14]. This important difference may be related to certain negative attitudes and beliefs among health professionals toward patients with diabetes in Latin America [53]. The presence of stigma in the health care context can have important consequences for users by limiting their access to diagnosis and treatment of the disease and to educational resources that are crucial for diabetes self-management.

The experience of stigma is not detached from the social situations experienced by individuals who are victims of it, for example, their social position [54]. The degree to which people internalize their social positions and the associated vulnerability have been related to feelings of inferiority, which can increase individuals’ likelihood of experiencing stigma [54]. Thus, people with diabetes who are also in a disadvantaged social position may be more sensitive to events involving stigma and discrimination toward them.

We also found a negative relationship between the stigma surrounding diabetes and the time elapsed since diagnosis. This association suggests that, as people live longer with the disease, they develop some resistance to stigma. Stigma resistance is an active process involving the use of one’s own experiences, knowledge, and skills to fight this phenomenon [55]. Two aspects that may contribute to stigma resistance are the development of more effective coping mechanisms and normative beliefs that naturalize the disease at certain life stages [56,57], both of which occur with greater probability in people who have been diagnosed with type 2 diabetes for a longer time.

The findings of this study support a negative relationship between stigma, self-esteem, and self-efficacy. In addition, these variables showed a significant mediating role in the relationship between stigma and self-management behavior, a result that is consistent with previous findings in patients with diabetes [23]. Corrigan, et al. [58] noted that this may be due to a process in which patients internalize the stereotypes that exist around their condition (e.g., those that allude to individual responsibility) and then fall into a state of personal devaluation that damages their self-esteem and self-efficacy and ultimately prevents them from achieving their goals (e.g., managing their disease).

Another relevant element is the association between perceived stigma and relationships with health care providers. Good relationships with providers is crucial to achieve adherence to self-management behaviors and better disease control [28]. Our findings support the significant mediating role of this variable in the relationship between stigma and self-management, indicating that greater perceived stigma leads to less involvement with health care providers and probably less trust in them, which may impair patients’ abilities to adhere to the behaviors necessary to control the disease. However, the cross-sectional nature of this study precludes determination of the direction of this association with any certainty.

Our results also show that higher levels of stigma are related to a higher level of diabetes-related stress, a finding that is consistent with previous research results [19,20,21,26,31]. However, we did not find a mediating effect of this variable in the relationship between stigma and self-management, differing from other similar studies [20,26,31]. Some researchers have noted that stress may be specifically related to certain self-management behaviors (for example, adherence to treatment) and not to a broader construct that includes several behaviors simultaneously, such as the one measured in this study [22]. Additionally, the involvement of other variables in this relationship cannot be ruled out. Snoek, et al. [59] pointed out that the effect of stress on self-management may be mediated by the presence of depressive symptoms, which may indicate the presence of a multiple mediation mechanism.

### Implications and Limitations

This study thoroughly explores the stigma surrounding diabetes and its associations with psychosocial and behavioral variables in a population in Latin America, where evidence on this topic is scarce.

Our results have important implications for health care providers as the health care setting seems to be a more relevant source of stigmatization for people with type 2 diabetes than other settings, such as the family or work environment. Health care providers should be familiar with the relevance of stigma for this group and develop skills to maintain an effective relationship free of judgment and focused on patients’ needs [60]. In this line, language is fundamental. The American Diabetes Association has recommended that health care providers use neutral, stigma-free, inclusive, and person-centered language [61]. Words such as “noncompliant”, “uncontrolled”, “unmotivated”, and “diabetic” (as an adjective) should be avoided as labeling people with diabetes in this manner constitutes a stigmatizing practice.

Health care providers can also implement interventions aimed at strengthening certain coping mechanisms in people with diabetes to combat the effects of stigma. Our results provide guidance in this respect. People who have been recently diagnosed and those with a low socioeconomic status are particularly vulnerable to stigma and would thus benefit more from such interventions. The identification of mechanisms involved in the relationship between stigma and self-management also provides clues about possible interventions to confront this phenomenon. Strategies that focus on improving self-esteem and self-efficacy as coping mechanisms (e.g., psychoeducation, motivational interview) have shown benefits in other stigmatized populations [3,62], and their implementation in patients with diabetes appears to be promising.

Another aspect to consider is the group or individual format of the intervention. Some systematic reviews have shown evidence of the effectiveness of group interventions to combat stigma [62,63]. Group support seems to be a fundamental element, and even a recent randomized control study showed that, regardless of the intervention’s content, the group context would reduce the levels of stigma [64]. This finding is consistent with the literature in psychology, which suggests that group membership is associated with more resistance to stigma, perception of control over one’s life, and self-esteem [65,66].

From a more structural standpoint, mass communication campaigns could be implemented to combat the stereotypes associated with type 2 diabetes, or institutional policies aimed at avoiding all forms of discrimination against people with the disease can be established. To continue achieving progress on this topic, further research must be increased in other Latin American countries. Existing stereotypes about diabetes and the prevalence of stigma in other countries and its associations with other variables such as access to health care still require investigation in the region.

However, the study has some limitations. This is a cross-sectional study; therefore, causal relationships between the study variables cannot be inferred, the sample was not probabilistic, and the stigma process may change in other cultural settings (e.g., different countries). These limitations justify the need for further studies in the region to gather evidence on this topic and to explore more complex relationships that simultaneously consider the roles of different mediators involved in the effects of stigma.

## 5. Conclusions

The perception of stigma was frequent in the analyzed sample, and health care represented a relevant source of stigmatization for these patients. Regarding the relationship between stigma sociodemographic and clinical factors, patients diagnosed with type 2 diabetes more recently, and patients with low socioeconomic status were particularly vulnerable to this phenomenon. Additionally, stigma had a relationship with different psychosocial variables. A greater perception of stigma was associated with lower self-efficacy, self-esteem, self-management skills, and satisfaction with health care and higher levels of stress. Self-efficacy, self-esteem, and satisfaction with health care had a mediation role in the relationship between stigma and self-management behaviors. These variables would be part of the mechanisms through which the perception of stigma harms self-management behaviors. More studies about these mechanisms and interventions related to stigma surrounding diabetes are crucial in the Latin American region.

## Figures and Tables

**Figure 1 ijerph-18-12657-f001:**
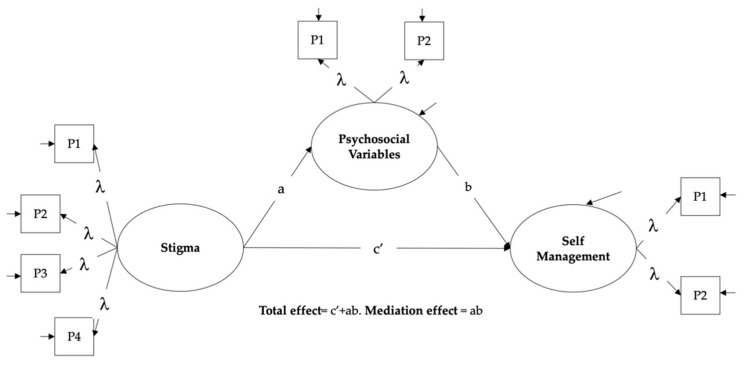
Structural Equation Model for Mediation Analysis. The paths a, b, and c’ represent regression coefficients. Circles: latent variables. Squares: indicators, parcels (P) in this case. λ: factor loadings.

**Figure 2 ijerph-18-12657-f002:**
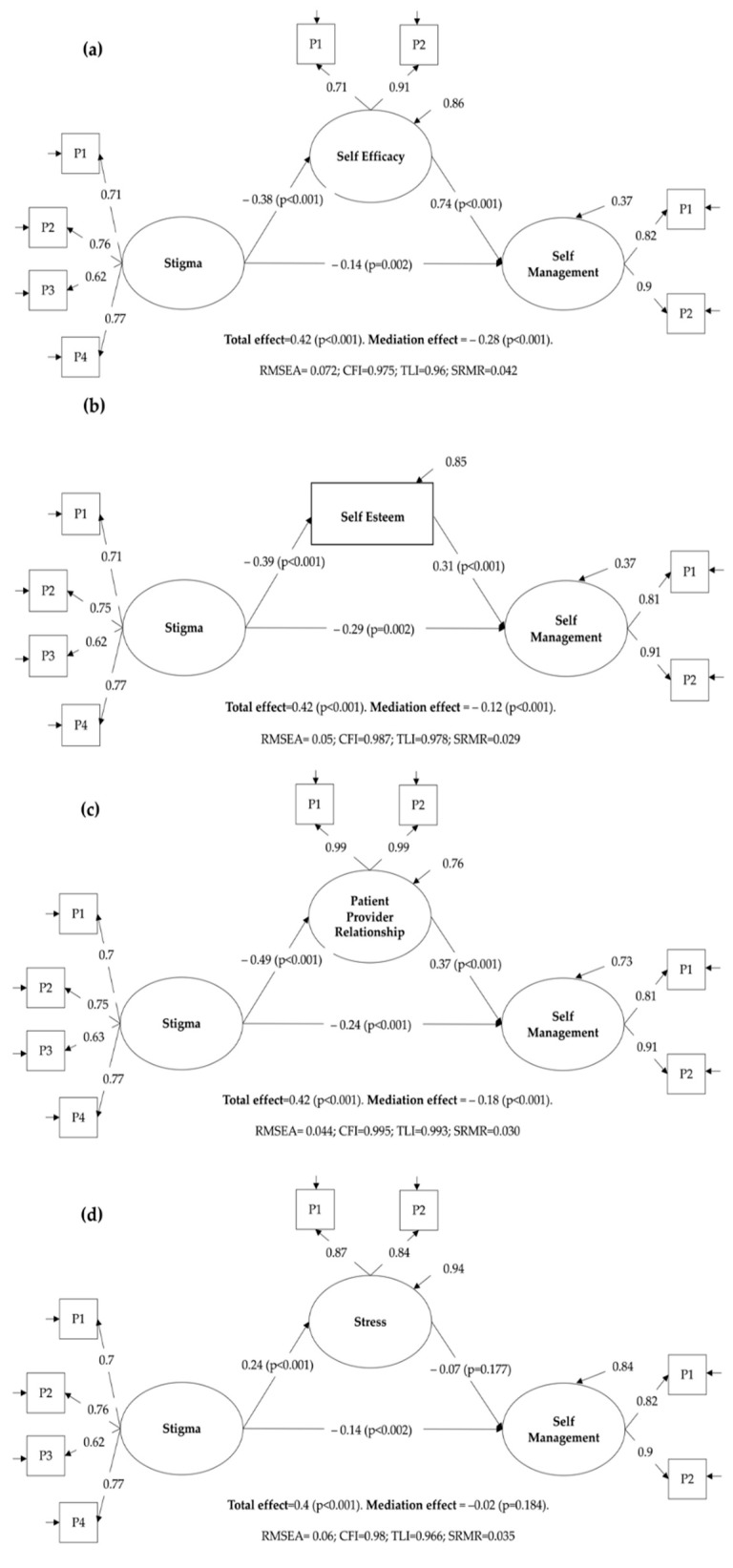
Structural equation models. The panels show different mediation models for the relationship between stigma and self-management. The mediators in different panels are: (**a**) self-efficacy, (**b**) self-esteem, (**c**) patient provider relationship, and (**d**) stress. The fit statistics and mediation effect are specified in each panel. P: parcel, RMSEA: Root Mean Square Error of Approximation, CFI: Comparative Fit Index (CFI), TLI: Tucker Lewis Index, and SRMR: Standardized Root Mean Square Residual.

**Table 1 ijerph-18-12657-t001:** Sociodemographic and clinical characteristics of participants (n = 501).

Participant Characteristic		N (%) or Mean (SD)
Age (years)		60 (12)
Subjective Socioeconomic Status		3.73 (1.22)
Sex	Men	184 (36.7%)
	Women	317 (63.3%)
Time since diabetes diagnosis (years)		8.78 (8.14)
Diabetes-related complications	Yes	130 (73.9%)
	No	368 (26.1%)
Use of Insulin	Yes	178 (35.5%)
	No	323 (65.5%)

**Table 2 ijerph-18-12657-t002:** General analysis for psychosocial and self-management variables.

Variable	Descriptive Statistics					Reliability	Correlations				
	Mean	SD	Min	Max	Missing	α	1	2	3	4	5
1. Stigma	49.79	7.11	25	74	6	0.72	1				
2. Self-Esteem	2.93	0.91	0	4	1	-	−0.36	1			
3.Stress	4.88	2.82	0	16	1	0.75	0.20	−0.28	1		
4. Self-Efficacy	43.19	10.07	10	65	0	0.81	−0.37	0.42	−0.23	1	
5. Patient-provider relationship	30.66	8.11	5	50	0	0.99	−0.45	0.37	−0.15	0.56	1
6. Self-Management	16.76	3.66	9	29	3	0.65	−0.24	0.35	**−0.06**	0.51	0.30

α: Cronbach’s alpha. In bold non significative correlations.

**Table 3 ijerph-18-12657-t003:** Item analysis for DSAS-2.

Item	Strongly Disagree		Disagree		Unsure		Agree		Strongly Agree	
	N	%	N	%	N	%	N	%	N	%
Some people think I cannot fulfill my responsibilities (e.g., work, family) because I have type 2 diabetes	32	6.4%	302	60.3%	1	0.2%	160	31.9%	6	1.2%
Because of my type 2 diabetes, health professionals have made negative judgments about me	18	3.6%	178	35.5%	-	-	268	53.5%	37	7.4%
Because I have type 2 diabetes, some people assume I must be overweight, or have been in the past	12	2.4%	193	38.6%	1	0.2%	201	40.2%	93	18.6%
Some people treat me like I’m “sick” or “ill” because I have type 2 diabetes	35	7.0%	226	45.1%	-	-	230	45.9%	10	2.0%
There is blame and shame surrounding type 2 diabetes	19	3.8%	462	92.2%	1	0.2%	19	3.8%	-	-
I feel embarrassed in social situations because of my type 2 diabetes	21	4.2%	455	90.8%	2	0.4%	22	4.4%	1	0.2%
I have been discriminated against in the workplace because of my type 2 diabetes	31	6.2%	431	86.2%	2	0.4%	34	6.8%	2	0.4%
Health professionals think that people with type 2 diabetes don’t know how to take care of themselves	17	3.4%	40	8.0%	-	-	303	60.8%	138	27.7%
I’m ashamed of having type 2 diabetes	57	11.4%	421	84.0%	2	0.4%	21	4.2%	-	-
Some people see me as a lesser person because I have type 2 diabetes	360	71.9%	131	26.1%	3	0.6%	7	1.4%	-	-
Because I have type 2 diabetes, I feel like I am not good enough	391	78.0%	104	20.8%	1	0.2%	5	1.0%	-	-
There is a negative stigma about type 2 diabetes being a “lifestyle disease”	8	1.6%	53	10.6%	21	4.2%	382	76.4%	36	7.2%
Having type 2 diabetes makes me feel like a failure	170	33.9%	330	65.9%	1	0.2%	-	-	-	-
Some people exclude me from social occasions that involve food/drink they think I shouldn’t have	24	4.8%	337	67.5%	2	0.4%	117	23.4%	19	3.8%
I feel guilty for having type 2 diabetes	17	3.4%	274	54.8%	6	1.2%	196	39.2%	7	1.4%
I have been told that I brought my type 2 diabetes on myself	11	2.2%	166	33.3%	2	0.4%	231	46.3%	89	17.8%
I have been rejected by others (e.g., friends, colleagues, romantic partners) because of my type 2 diabetes	16	3.2%	460	91.8%	-	-	24	4.8%	1	0.2%
I blame myself for having type 2 diabetes	17	3.4%	244	48.8%	14	2.8%	219	43.8%	6	1.2%
Because I have type 2 diabetes, some people judge me for my food choices	6	1.2%	91	18.2%	3	0.6%	308	61.5%	93	18.6%

**Table 4 ijerph-18-12657-t004:** Regression analysis for the relationship between stigma surrounding diabetes, sociodemographic, and clinical variables.

Variable	Model 1		Model 2	
	β	*p* Value	β	*p* Value
Age	−0.13	0.013	−0.08	0.138
Sex	0.07	0.171	0.06	0.198
Subjective Socioeconomic status	−0.15	0.001	−0.13	0.002
Use of Insulin			−0.06	0.186
Time since diabetes diagnosis			−0.23	0.001
Diabetes-related complications			0.04	0.379

β: Standardized regression coefficient.

**Table 5 ijerph-18-12657-t005:** Regression analysis for the relationship between stigma surrounding diabetes, sociodemographic, and clinical variables.

	Self-Efficacy (N = 490)		Self-Esteem (N = 490)		Stress (N = 489)		Patient-Provider Relationship (N = 489)		Self-Management (N = 490)	
	β	*p* Value	β	*p* Value	β	*p* Value	β	*p* Value	β	*p* Value
Age	0.04	0.33	0.01	0.79	−0.03	0.55	0.02	0.65	0.07	0.14
Sex	0.03	0.45	−0.09	0.02	−0.16	<0.001	0.10	0.01	0.06	0.19
Subjective Socioeconomic status	0.14	0.00	0.08	0.05	−0.04	0.32	−0.05	0.19	−0.10	0.03
Use of Insulin	−0.05	0.23	−0.06	0.22	0.12	0.02	−0.07	0.08	0.01	0.86
Time since diabetes diagnosis	−0.11	0.01	−0.15	<0.001	0.03	0.49	−0.05	0.32	0.03	0.46
Diabetes-related complications	0.06	0.18	0.04	0.36	0.11	0.02	0.01	0.84	−0.12	0.01
Stigma	−0.36	<0.001	−0.37	<0.001	0.23	<0.001	−0.46	<0.001	0.22	<0.001

β: Standardized regression coefficient.

## Data Availability

The data presented in this study are available on request from the corresponding author.

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
