# Peer review of "A Cross-Sectional Analysis of the Stigma Surrounding Type 2 Diabetes in Colombia"

_ijerph, 2021, doi:10.3390/ijerph182312657_

Round 1

Reviewer 1 Report

In this paper, the authors analyze the level of stigma surrounding type 2 diabetes in the Colombian population and its relationships with sociodemographic, clinical, psychosocial variables and behaviours related to the management of the disease (self-management behaviours).

The authors selected to investigate a challenging issue with a significant impact on people all over the world. Its structure is acceptable and meets the specifications of the article.
1)Starting from the title of the paper, it is too long and should be reduced. A recommended title is: "A cross-sectional analysis of the Stigma Surrounding Type 2 Diabetes in Colombia".
2)The abstract describes the contribution of the work and gives details about the results of Type 2 diabetes stigma analysis. In the abstract, the authors should emphasize the background and the motivation of their research.
3)The introduction is well written, but the title of subsection 1.1 could be omitted and, its contents merged with the introduction.
Lines 114-121 needs appropriate referencing.
At the end of the introduction, you should mention the structure of the paper.
4)The introduction could be further enriched. The authors could provide more technical references regarding the prediction of diabetes.
Such reports are i)Machine Learning Tools for Long-Term Type 2 Diabetes Risk Prediction. doi 10.1109/ACCESS.2021.3098691
ii)Prediction of Type 2 Diabetes using Machine Learning Classification Methods. doi https://doi.org/10.1016/j.procs.2020.03.336
5)The final sample size of this study was 501 participants.
Ηow it turns out that the relationships between stigma and psychological variables were moderate (β= |0.39|)?
Could you document and analyze more lines 156-161?
6)In the subsections of section 2, the patients answered 5 or 7 items of the Spanish version of the self-efficacy scale for diabetes. Could you indicate in the text what these items are?
7)Subsection 2.3 analysis needs reinforcement.
Could you give more information about the models you used by citing the appropriate references?
8)I suggest the authors represent the results in figures.
9)The system specifications used in the experiments should be mentioned. Please, demonstrate the environment of the experiments in detail. 
10)The conclusion section could be further improved to elaborate on and highlight the results found in the simulation models.
11)Conclusions could discuss future research directions and extensions of this work.

Reviewer 2 Report

Thank you very much for this manuscript. This topic is very interesting and in the view of percentage of people with diabetes is very actual. Thank authors for hard work on this manuscript. I have only some recommendation:

1) Abstract is clear and adequate. Maybe it can be shorter.

2) Introduction is adequate and correspondence with aim of article. Maybe it is for thinking if it is necessary to separate subchapter. I think that these information complete information in introduction. Also it can be better to connect or to intertwine text. General information about stigma can be shorter. Some information from this part can be used in discussion. 

3) Materials and methods need more attention. It seems to little bit unclear. It can help to better specify name of subchapters.

4) In part of results it is necessary to add explanation of abbreviations which are used in tables. 

5) Discussion is very interesting for me. Thank you for your work.

Reviewer 3 Report

The work is well planned but there are some shortcomings that hinder its compressive reading, so I recommend the authors:

The abstract should show the questions used to carry out the study.

In the results of the summary, some more in-depth detail should be made.

The introduction of the work must be reformulated, the contents of section 1.1 are very interesting, but I consider that they are part of a background  of the specialists to whom this work is addressed, so I recommend the authors to synthesize some sentences these aspects. On the other hand, the last paragraph should be deleted and replaced by the objective of the work, in a clear direct and concise way.

section 2.1 should be reformulated in a clearer and more concise manner that addresses only recruitment, inclusion/exclusion criteria, description of the population and addresses ethical issues.

The discussion should contain more reflections from the authors of the interventions that could improve this situation.

The conclusions do not respond directly to the objective set.

Round 2

Reviewer 1 Report

I have no additional remarks on the revised version.

Author Response

Thank you.  

Reviewer 3 Report

The authors did a great job trying to improve the manuscript, but they did not correctly address the comments corresponding to section 1.1 and 2.1.
